# Natural or Human Landscape Beauty? Quantifying Aesthetic Experience at Longji Terraces Through Eye-Tracking

**DOI:** 10.3390/jemr18030015

**Published:** 2025-05-07

**Authors:** Ting Zhang, Yue Jiang, Donghong Liu, Shijie Zeng, Pengjin Sheng

**Affiliations:** 1College of Tourism & Landscape Architecture, Guilin University of Technology, Guilin 541004, China; 15116612918@163.com (Y.J.); 13398493137@163.com (S.Z.); shengpengjin888@163.com (P.S.); 2Guangxi Natural Resources Vocational and Technical College, Nanning 530000, China

**Keywords:** aesthetic experience, Longji Terrace landscape, SOR modeling, eye movement experiments

## Abstract

This study investigated tourists’ visual perception, aesthetic experience, and behavioral intentions across four types of landscapes. A total of 353 questionnaires were distributed on-site, and the SOR model was used to examine the visual stimuli and aesthetic responses perceived by tourists, followed by laboratory eye-tracking to observe tourists’ points of attention on the Longji Terraced Fields landscape Key findings reveal that different residences and revisiting conditions affect tourists’ visual attention, with the most attention given at the intersections of landscape elements. Furthermore, although landscape visual stimuli do not significantly affect the intention response, eye movement parameters are positively correlated with aesthetic experience. The study contributes to understanding tourist aesthetic perception in terraced rice field landscapes and provides Chinese cases for the aesthetic appreciation of the terrace landscape.

## 1. Introduction

The report of the 20th CPC National Congress clearly puts forward that in order to promote “Beautiful China”, it is necessary to adhere to the principle that “green water and green mountains are golden silver mountains” On this basis, the great significance of the construction of “Beautiful China” is put forward [1]. In terrace landscape tourism, the beauty of the architectural landscape, the ecological beauty, the cultural beauty, and the agricultural beauty are intertwined with each other, resulting in rich and diverse aesthetic values. With the continuous growth of the economy and the enrichment of people’s travel experience, the aesthetic experience becomes more and more a key factor in determining people’s willingness to travel, and the phenomenon of poor post-travel experience, poor impression, low satisfaction, and low revisit rate of the travelers can be properly explained. This study proposes the direction of the research on the tourism experience towards the aesthetic ecosystem and the conception of the research and provides new perspectives and ideas for solving such problems.

The Longji Terrace can be called the original hometown of the world’s terrace. It is a unique farmland landscape in the Longji Mountain area of Guangxi Zhuang Autonomous Region, China, and has been included in the World Cultural Heritage. Longji terraces are located in Longji Mountain, located in Pingan Village, Longji Town, Longsheng Autonomous County, Guilin City, Guangxi Zhuang Autonomous Region, between east longitude 110°4′06″~110°11′52″ and north latitude 25°42′33″~25°50′15″. As early as 6000 to 12,000 years ago, primitive cultivated japonica rice was one of the birthplaces of artificial cultivated rice in the world. It belongs to the Nanling mountain across the city ridge, rolling mountains, overlapping peaks, and vigorous momentum. Longji terraces also show the phenomenon of staggered terraces and forests. Under a long terrace, there is often a forest, and such a layout plays a good role in soil and water conservation. With its unique and rich geographical background, it is not only a magnificent natural landscape but also an important part of the Chinese traditional farming culture, as shown in Figure 1.

As an important tourism resource, the terraced landscape needs scientific planning and layout. Landscape visual quality evaluation, as an important way of tourism resources evaluation, uses qualitative and quantitative methods to determine the visual demand for the landscape [2]. At present, eye-tracking instruments are used in landscape visual quality evaluation, on the one hand, researchers use landscape type as an entry point, such as Zhao Ying et al. divided Tangjia Ancient Town in Zhuhai into four types of landscapes: historical and cultural, cultural creativity, modern facilities, and fresco graffiti, and then explored the differences in visual behavior in these four types of landscapes through eye-tracking experiments, respectively [3]. Guo Suling et al. conducted eye movement experiments on Hongcun from the aspects of settlement landscape, ecological landscape, cultural landscape, and agricultural landscape, and according to the different types of landscapes, the eye movement data and heat maps they obtained also presented diverse visual characteristics [4]. Su Si-Qing et al. explored the impact on the lived experience of natural landscape-based tourism by categorizing online reviews [5]. Yu Gao et al. used eye-tracking technology to analyze the characteristics and differences of visual behavior in different types of forest landscape spaces [6]. On the other hand, scholars have explored the influence of group characteristics on landscape evaluation, e.g., Ren divided Chinese tourists and British tourists into two groups, studied and explored the differences in their respective perceptions of landscapes, and came up with cross-cultural group landscape preferences, with British tourists preferring the colors of landscapes, and Chinese tourists preferring plant-based landscapes [7]. Wang Junyi et al. explored the differences in spatial perceptions of map symbols among students from different fields in terms of their gender, disciplinary background, and professional background [8]. Minanshu Sinha et al. explored how the gender selection of advertising spokespeople for fast-moving consumer goods (FMCG) can maximize advertising effectiveness.

At present, for the landscape visual quality evaluation system, subjective dimensions such as questionnaires and scoring methods are commonly employed to construct the evaluation system, and objective dimensions need further enrichment [9]. With the continuous advancement of eye-tracking equipment, Individual objective perception responses can be introduced into the assessment of landscape visual quality [4]. The eye movement analysis method currently focuses on the study of people’s visual perception of landscapes. Domestic eye movement used in landscape research mainly includes cultural landscapes [10], acoustic landscapes [11,12], visual landscapes [13], and forest park landscapes [14,15], as well as the visual evaluation of tourist destination landscapes [16]. The Xuanwu Lake Park and the Purple Mountain National Forest Park in Nanjing were selected to explore whether a single landscape element has an impact on the landscape vision. However, it is found that leisure furniture can significantly enhance the aesthetic evaluation and sensory experience of tourists in the Xuanwu Lake National Forest Park. However, the sub-optimal leisure furniture in Purple Mountain National Forest Park (Purple Mountain National Forest Park) has a negative impact on visual attractiveness and senses [17]. Overseas research on eye movement analysis is earlier, and the results of research on landscape using an eye-tracker are richer. For example, Nordh et al. studied the eye movement data of different landscape elements [18]. Gao et al. conducted a classification study on forest landscapes and found that people generally preferred dynamic water landscapes [6]. Yiping Liu et al. evaluated the audio-visual interaction effects of forest landscapes using eye tracking, and the results indicated that different types of forest landscapes affect visitors’ perceptions [19]. Xingcan Zhou et al. utilized eye-tracking technology to explore the differences in visual behavior related to landscape elements in urban lakeside parks and found a greater focus on artificial elements [20]. Chang Li et al. measured seven levels of forest light exposure using eye-tracking technology in virtual reality to explore the effects of different brightness levels on forest visual perception [21].

Currently, terraced landscapes are facing important challenges, and achieving multifunctionality in high-quality terraced landscapes as well as sustainable management requires solutions to provide scientific and technological support for the construction of the national ecological civilization and the revitalization of the countryside [22]. Therefore, exploring different landscapes through visual quality evaluation methods is the focus of this study. Among the numerous discussions on landscape evaluation, a popular approach proposed by the psychological school is to validate the “landscape-aesthetic” through the “stimulus–response” paradigm. The more common is the “stimulus–organism–response” model (SOR model), in which external factors are the stimulus, the individual’s physiological and psychological perception is the organism, and the individual’s psychology and behavior are the response [23]. However, landscape visual quality evaluations typically reflect only subjective perception and need to be supported by objective data [24].

In view of this, this study aims to reflect the objective perception of the participants through eye movement experiments, which greatly enriches the objective dimension in the visual quality evaluation system. Meanwhile, the current research on eye movement experiments in tourism landscapes does not involve terraced landscapes. To synthesize the aforementioned theoretical and contextual background, this study selected the Longji Terraces in Guangxi—a UNESCO-recognized agricultural heritage site characterized by diverse landscape typologies—as a case study. We employed a mixed-methods approach integrating eye-tracking technology and questionnaire surveys to systematically investigate tourists’ landscape aesthetic experiences. The findings offer practical insights for optimizing tourism destination management strategies to enhance visitor attraction and heritage conservation in terraced field landscapes.

## 2. Theory and Model Assumptions

### 2.1. Theories

Dual-systems theory mainly points out that there are two different ways of information processing in the human cognitive system, and its core lies in revealing the psychological mechanism of the human cognitive process. Based on the research by Jun and Vogt, the heuristic-system model from an interactive perspective can provide theoretical support for explaining tourists’ aesthetic experiences [25]. SOR theory states that environmental stimuli refer to external factors that may cause changes in cognition and mood [3]. At the same time, Donovan et al. stated that an individual’s mood and cognitive state shifting towards negativity or not can also affect his or her response to environmental stimuli. Together, these factors affect the individual’s perception and evaluation of aesthetic experiences [26]. More and more fields have begun to introduce the SOR model to investigate the role of controlled experimental design in influencing tourists’ experiences. SOR theory can be used in studying aesthetic experiences. For example, Zhao Ying et al. (2020) used the SOR theory to analyze eye movement behavior to improve the evaluation of landscape visual quality [3]. For example, based on the SOR theory, Deng Weiwei et al. (2021) proposed the application of the SOR theory to study the mechanism of interaction between aesthetic emotions and visual attractiveness [27].

### 2.2. Model Assumptions

Overseas scholars have explored the research objects and experimental methodologies from the aspect of tourism landscape influencing factors and evaluation research. Liu believes that subjects are more interested in natural elements, social interactions, etc. [28]. Potocka found that in most lake landscapes, the most dominant element is the tourists on the shoreline of the lake [29]. Liu and Carles studied tourists’ audio-visual interaction evaluations of forest landscapes and concluded that the aesthetic experience is improved when the soundscape is consistent [19,30]. Kong conducted a study on tourism advertisements and found that tourists’ experience while browsing a tourism website is the main factor influencing their attention and memory of tourism advertisements [31]. Thus, Hypothesis H2 is proposed: Landscape visual stimulation positively affects tourists’ aesthetic experience.

Muñoz-Leiva used consumers’ memory of banner advertisements as a determinant of advertising effectiveness and verified that the number of gazes and duration of visits had a positive effect on consumers’ memory [32]. Lourencão explored tourism symbols and slogans of tourist destinations and found that they did not influence tourists’ perceived effectiveness of advertisements [33]. From this, it is proposed: H1: Landscape visual stimuli positively influence tourists’ intended responses.

At present, researchers have recognized the importance of aesthetic experience in the development of tourism and have also carried out relevant empirical studies. The research on aesthetic experience is mainly focused on the aesthetic senses, aesthetic emotions, and the aesthetic value of tourism experience, and Zhang et al.’s study examines the influence of aesthetic elements on tourists’ aesthetic emotions and aesthetic judgment [34]. Deng Weiwei et al. believe different landscape types and photographic aesthetics have a positive impact on tourists’ aesthetic emotions and visual appeal [27]. Therefore, Hypothesis H3 is proposed: Aesthetic experience positively influences tourists’ intention response. Few scholars have explored the comprehensive experience of tourists from the perspective of aesthetics, using pictures as stimulus materials. Jiang Liao et al. explored the impact of tourism aesthetic experience on aesthetic consumption [35]. Breiby et al. concluded that aesthetic experience has a certain impact on satisfaction and loyalty [36]. At the same time, aesthetic experience also has an impact on the overall evaluation of tourists. Thus, Hypothesis H4 is proposed: Aesthetic experience mediates between landscape visual stimuli and intended responses.

In summary, the purpose of this study is fourfold: ① We aimed to investigate whether landscape visual stimuli directly and positively influence tourists’ intention responses. ② Whether landscape visual stimuli positively influence tourists’ aesthetic experience. ③ Aesthetic experience positively influences tourists’ intentional responses. ④ We investigated whether aesthetic experience has a mediating role between landscape visual stimuli and intentional responses. The proposed hypothesized mediation model is shown in Figure 2.

## 3. Research Methods

### 3.1. Research Subjects

On-site and laboratory eye-tracking experimental methods were used for tourists and college students at the Longji Terraces, respectively. In the on-site survey at Longji Terraces, there were 144 males and 209 females; 210 participants’ permanent residence was in towns and cities; the educational background was mostly undergraduate and junior college; the ethnicity was mainly Han Chinese, and there were 91 Zhuang among the minorities; tourists who loved the ethnic and cultural landscapes amounted to 19; the largest number of them came from Guangxi, and 169 of them were first-time visitors to Longji Terraces; the number of tourists in the age groups of 17–28 years old and 29–50 years old who were willing to participate in the experiment was high; the majority of them have an income of 2001–5000 yuan. In the laboratory experiment, there were 25 people who lived in cities and towns; most of the tourists had educational backgrounds of undergraduate and graduate students or above; the ethnic groups were mainly Han Chinese, and there were 10 in the Zhuang ethnic group among the ethnic minorities; 19 tourists were fond of ecological landscapes; most participants came from Guangxi, and among them, 21 people were first-time visitors to Longji Terraced Fields; more tourists were willing to take part in the experiment at the age stages of 17–28 and 29–50; and most tourists had incomes of 2001–5000 yuan. The majority of the tourists were from Guangxi; 21 of them visited Longji Terraces for the first time; more tourists in the age groups of 17–28 and 29–50 were willing to participate in the experiment; and the majority of them had an income of 2001–5000 yuan.

### 3.2. Research Design

#### 3.2.1. Experimental Preparation

##### Selection of Landscape Pictures

Due to the limitations of human and financial resources and other factors, pictures were chosen as the medium for on-site research. The research materials were selected from the actual photographs uploaded by tourists in the mainstream travel media featuring the Longji Terraced Landscape. On the basis of selecting images, this study defined categories for the Longji Terraces including settlement and architectural landscape, ecological landscape, ethnic cultural landscape, and agricultural landscape, identified representative elements of each thematic landscape, standardized the pixel size of the images to be 1600 × 900, and solicited the opinions of a number of experts, Finally, for the field experiments, 10 images of each landscape type were selected from 100 images, totaling 40 representative images as experimental stimuli; for the laboratory experiment, 3 images of each landscape type were selected, totaling 12 representative images as experimental stimuli, see Appendix A for the selected original image. To account for primacy effects, 4 additional pictures were selected from the remaining images, one for each landscape type, to serve as warm-up stimuli

##### Subject Selection

This study was conducted as a pre-study on 14 May 2022, with a total of 11 subjects participating in the experiment. By analyzing the data, the problems of the eye movement experiment were summarized, and the formal experiment was improved to obtain reliable statistical results. Since most of the previous eye movement studies selected a relatively homogeneous group of college students, in view of this, this study selected case land tourists as subjects, which enriched the type of subject group to a certain extent. Considering the issues of accessibility, efficiency, cost, and the limitations of field experimental conditions. Fifty-eight participants (30 females, 28 males) at the Longji Terraces site were invited to participate in the experiment, and the effective sample size was 45 (23 females, 22 males) after excluding subjects with insufficient eye movement sampling rate (<50%), whose naked and corrected visual acuities fell within the relatively normal range of 0.8 to 1.5. Participants in the laboratory experiment were undergraduate and graduate students from Guilin University of Technology, among others, with mild myopia of 400 degrees or less. Fifty-five participants (29 females, 26 males) were invited to participate in the experiment, and the effective sample size was 45 (25 females, 20 males) after excluding subjects with a low eye movement sampling rate (<50%).

##### Equipment Selection

The Tobii Glasses3 eye-tracking device was used to collect eye-movement data as an auxiliary tool for vision research and as an effective quantitative index, and a Lenovo Xiaoxin 13-pro laptop (13.3 inches, screen resolution 1600 × 900) was used for stimulus presentation in the eye-movement experiments and to analyze the data at a later stage. In addition, an iPad was prepared for the subjects to browse the pictures when filling out the questionnaire. During the experiment, the sampling rate of the eye-tracking device was adjusted to 250 Hz.

##### Scale Selection

Three scales were employed for aesthetic experience: an aesthetic sensory scale, an aesthetic emotion scale, and an aesthetic spirit scale, in which the aesthetic sensory scale was adapted from Long Yalin et al.’s (2020) research on aesthetic senses [37] and combined with Fan Yaming et al.’s (2022) research on the landscape elements of the Longji Terraced Fields, being modified accordingly based on the elements of the aesthetic senses and the four landscape types in the Longji Terraced Fields, namely settlement and architecture, ecology, ethnic culture and agriculture [2]. The scale design of aesthetic emotion was mainly based on Deng Weiwei et al.’s (2021) study of aesthetic emotion [27] and modified accordingly based on the characteristics of the Longji Terraced Landscape. The scale design of aesthetic spirit mainly draws on Long Yalin et al.’s (2020) study on the connotation of aesthetic spirit [37] and combines with Bao Qingqing et al.’s (2021) study on the connotation of the genetic characteristics of the landscape of Longji Terraced Fields [38], and modifies it accordingly.

Intentional responses synthesized the previous research results, and the four important intentional responses of comprehensive experience, overall impression, satisfaction, and willingness to go out or revisit were selected for in-depth study [3].

#### 3.2.2. Eye Movement Indicator Selection

At the end of the experiment, considering that each participant’s test was distributed in different time periods, the eye movement experiment was carried out in a natural environment, and several eye movement metrics were selected to quantify the eye movement data. Specifically, the eye fixation frequency, the gaze frequency, the average number of fixations, and the average fixation duration were selected as the indexes [3]. The eye-beat frequency indicates the attractiveness of the landscape to the participants; the gaze frequency indicates the information perception efficiency of the landscape visual stimuli; the average number of gaze times indicates the degree of the landscape visual stimuli; and the average gaze time indicates the participants’ cognitive effort on the landscape.

### 3.3. Research Process

Firstly, the experimental personnel introduced the experimental process as well as the requirements to the participants and assisted the visitors in understanding the notification of the eye movement experiment. After the participants took a short rest, they helped them to wear the eye-tracking device and guided the participants to sit in front of the picture display screen at about 65 cm away, and then the experimental personnel assisted the subjects to calibrate the eye-tracking device. Once the calibration was successful, the experiment began. The participants viewed 40 pictures freely, while the eye-tracking device recorded the eye-tracking data of the subjects, and finally, a questionnaire was administered at the end of the eye-tracking experiment.

### 3.4. Experimental Data Processing

After the experiment, the data were processed with the help of Tobii Pro Lab software (Tobii Glasses 3). The required eye movement index data were extracted and imported into SPSS23.0 for statistical analysis. Incomplete data were deleted, and the number of participants with valid eye movement data was finally determined to be 45.

### 3.5. Data Analysis Methods

Eye movement data is an effective method for understanding people’s visual attention and emotion in behavior, and eye movement data can be used to understand the aesthetic and cognitive processes regarding landscapes. This article mainly analyzes the following 3 aspects. Firstly, the spatial data are visualized, and the eye movement heat map obtained from the experiment is analyzed to identify the attention points of tourists. Secondly, the eye movement experiments for different demographic groups are analyzed to explore the effects of 7 objective conditions, namely gender, permanent residence, educational background, ethnicity, number of revisits, age, and income level, as well as sampling rate, on the 4 eye movement indicators in terms of the average number of the average number of fixations, the average fixation duration, the fixation frequency and the frequency of gaze. The significance of these different eye movement indicators in terms of different demographic groups is investigated. Finally, the effect of different eye movement data is explored in the following 3 aspects. The significance of these indicators was investigated; finally, the effects of different landscape types on eye movement behaviors were explored.

## 4. Results

### 4.1. Reliability and Validity Analysis

The reliability of the total scale is 0.937, and the reliability of each component scale is above 0.6; these data are acceptable. The KMO value of the scale is 0.818 > 0.5, and Bartlett’s spherical test reaches 0.000, which indicates that the scale has good validity. Overall, the questionnaire has good reliability and validity.

### 4.2. Factor Analysis

The Longji Terraces aesthetic experience, as well as the intention to respond to each item, was tested, and it was found that the items were suitable for further factor analysis (the KMO value of the aesthetic experience was 0.781, and the KMO value of the intention to respond was 0.708, and the significance level of both was 0.000), and the principal component analysis was applied to the factor analysis of the items of the aesthetic experience and the intention to respond sections of the questionnaire, and all the factors’ variance explanation rates were greater than 50%.

### 4.3. Correlation Analysis

In this study, the correlation analysis of participants’ aesthetic experience and intended response was conducted through the questionnaire. Table 1 shows that there is a significant positive correlation between the aesthetic senses, aesthetic emotions, and aesthetic spirit in the subjective evaluation, and the comprehensive experience, satisfaction, and overall impression in the intention response, and the intention to travel/revisit.

### 4.4. Regression Analysis

Taking the intention response as the dependent variable to be explained in this study, and taking the three dimensions of aesthetic sensation, aesthetic emotion, and aesthetic spirit in the aesthetic experience as the independent variables, regression analysis was conducted to obtain the results:Y = 1.003 + 0.506 × 1 + 0.275X2 − 0.013X3 (1)
where (X1 is aesthetic senses, X2 is aesthetic emotions, X3 is the aesthetic spirit). Through the above regression equation, it can be found that the three indicators of aesthetic senses, aesthetic emotions, and aesthetic spirit have a significant effect on tourists’ intention response. Through the above regression equation, it can be seen that the smaller the aesthetic spirit value is, and the larger the aesthetic senses and aesthetic emotions values are, the higher the tourists’ intention response will be. Overall, the higher the tourists’ aesthetic experience, the higher their intention response will be, which again verifies hypothesis H3.

### 4.5. Experimental Results

(1)Comparative analysis of visual attention based on eye movement heat maps

The color depth of the heat map represents the length of the average gaze time and the level of interest of the subjects on the map. The visualization of the location of the gaze points and the average gaze time of all the subjects was unfolded to obtain the heat map of the landscape. In this experiment, four dimensions, namely, settlement and architectural landscape, ecological landscape, ethno-cultural landscape, and agricultural landscape, were chosen to be experimented with in the field as well as in the laboratory to obtain the following heat maps (Figure 3 and Figure 4). It was found that the attention was more dispersed in the field experiment, and the subjects’ visual focus was relatively concentrated when observing the settlement and architectural landscapes, and their attention was focused on the buildings and their surrounding scenery. For both ecological and agricultural landscapes, the subjects did not focus their attention points on a specific element, but rather they looked across the field and the areas with big mountains. Among them, the ecological landscape’s attention points were distributed in a wider area and had a wider visual range [4]. However, the hot spot of attention of the ecological landscape with its single color was concentrated at the junction of elements. Ethnic and cultural landscapes had the most concentrated visual range, with attention points focused on clothing, and character areas, and concentrated in the near landscape. Agricultural landscapes, on the other hand, mainly focused on the figures of farming activities as well as farming tools. In the laboratory experiments, the distribution of attention points was relatively concentrated. When observing the settlement landscape, the attention points were mainly on the buildings and the surrounding scenery, and in the ethnic culture landscape, the attention points were spreading from the center to the surroundings, still focusing on the costumes and the figures.

(2)Comparative eye movement analysis in the field as well as in the laboratory for different demographic characteristics

In order to investigate the differences in visual preferences of different groups for the four landscape types, this study grouped the participants by eight criteria: gender, number of revisits, usual place of residence, ethnicity, income level, age, educational background, and sampling rate, and conducted one-way ANOVA on the data derived from four indicators: eye fixation frequency, gaze frequency, average number of fixations, and average fixation duration, respectively, in the laboratory and in the field, and the results of the laboratory experiments (See Appendix Table A1 and Table A2 for details) showed that: First, with the sampling rate as the control variable, there was a significant difference between average number of fixations and the eye fixation frequency, indicating that different sampling rates had a significant effect on the visual stimulation of the landscape, i.e., more than 91% of the subjects were more easily attracted by the landscape of the Longji Terraces. Second, using gender, number of revisits, usual residence, and professional background as control variables, the test results were not significantly different. The results of the field experiment showed that:

① With the usual place of residence as the control variable, there was a significant difference between average fixation duration and fixation frequency, indicating that the different places of usual residence had a significant effect on the visual stimulation of the landscape. There was a significant difference between subjects from rural areas and urban areas. The sig value of the average gaze duration in the usual place of residence was 0.016, and the average gaze duration of subjects from urban areas (4.845 times) was significantly lower than that of subjects from rural areas (6.414 times), which indicated that subjects from rural areas were more interested in the landscape of the Longji Terraces; the sig value of the eye hopping frequency in the usual place of residence was 0.039, and that of the urban area (4.797 times/s) was 0.039, and that of the urban area (4.797 times/s) was 0.039. (4.797 times/s) was significantly lower than that of subjects from rural areas (6.641 times/s), indicating that tourists from rural areas were more likely to be attracted to the landscape of Longji Terraces and could learn more information from it.

② Using whether or not to revisit as the control variable, the main effect concomitant probability value of whether tourists have had revisited experience is 0.02, which is less than 0.05, and there is a significant difference in whether tourists have had revisited experience. Specifically, the average number of gazes by tourists without revisiting experience (6.352) was significantly higher than that of tourists with revisiting experience (4.834), indicating that tourists visiting Longji Terraces for the first time paid more attention to the information in the pictures of the Longji Terraces’ landscape, and were more interested in the Longji Terraces’ landscape. The main effect concomitant probability value of whether the tourists have had revisited experience is 0.048, which is less than 0.05, and there is a significant difference in whether the tourists have had revisited experience or not. Specifically, the eye fixation rate of tourists who visited Longji Terraces for the first time (6.555 times/s) was significantly higher than that of tourists who had visited Longji Terraces before (4.795 times/s), indicating that tourists who visited Longji Terraces for the first time were more easily attracted to the landscape of the terraced rice fields, suggesting that first-time visitors to the terraced rice fields in Longji processed more visual information and had a higher level of interest in the landscape of the terraced rice fields in Longji.

③ Using gender, ethnicity, income, age, educational background, and sampling rate as control variables, the test results were not significantly different (sig > 0.05).

(3)Effects of different landscape types on eye movement behavior

This study explored the traditional landscape classification, referring to the standard “Classification, Investigation and Evaluation of Tourism Resources” (GB/T18972-2017 [39]) and related literature, and classified the stimulus materials into four types of landscapes: settlement and architectural landscapes, ecological landscapes, ethnic cultural landscapes, and agricultural landscapes.

First of all, this study used SPSS23.0 to put the testers’ eye fixation rate, fixation count, average fixation count, and average fixation duration of the four types of landscapes into a one-way ANOVA (Table 2). The eye movement experiments carried out in the laboratory found that there was no significant difference between the four (Table 3), the cognitive attention of the subjects was close to each other, and the ecological landscapes had the highest level of concern on the 2 eye movement metrics of the eye fixation rate and the average fixation count. In the eye movement experiments conducted in the field, there was no significant difference (sig > 0.05) in eye fixation rate, fixation count, average fixation count, and average fixation duration, indicating that for different types of landscapes, the subjects’ cognitive degree and attention were close to each other. On the three eye movement metrics of average fixation count, average fixation duration, and eye fixation rate, ecological landscapes had the highest attention, which can be further hypothesized that ecological landscapes seem to stimulate the interest of the testers more.

### 4.6. Hypothesized Model Analysis

In this study, the aesthetic experience of subjective evaluation, the intentional response, and the eye movement index of landscape visual stimulation were correlated in the laboratory as well as in the field, and the results of the study are shown in Table 4 and Table 5. The laboratory results showed that there was no significant correlation between the intentional response and the aesthetic experience of landscape visual stimulation, and therefore, hypotheses H1 and H2 were not valid. While the field experiments showed that there was no significant correlation between the intentional response in landscape visual stimuli with subjective evaluation, therefore, hypothesis H1 was not valid. Aesthetic emotion has a significant correlation with the average number of gazes and the frequency of eye fixations, and the correlation coefficients are 0.376 and 0.398, indicating that they are positively correlated. Therefore, hypothesis H2 is valid. Aesthetic sensation, aesthetic emotion, and aesthetic spirit in subjective evaluation have a significant correlation with comprehensive experience and satisfaction in intentional response, and aesthetic experience positively affects tourists’ intentional response. Therefore, hypothesis H3 was verified. Mediation effect test, using Bootstrap analysis, it can be seen that the influence path is landscape visual stimulation → aesthetic experience → intentional response, 95% confidence interval is −0.0014~0.0013, which contains 0, the mediation effect did not reach significance, the aesthetic experience does not have a mediating role in the landscape visual stimulation and intentional response relationship, so H4 was rejected.

## 5. Conclusion and Discussion

### 5.1. Conclusions of the Study

From the perspective of different types of Longji Terraces landscapes, this study draws the following conclusions by analyzing the tourists’ questionnaires and the responses of eye-movement data in the terraced landscape photographs:

① From the heat map analysis, it can be seen that different stimulus materials generally present two kinds of gaze characteristics, one is that the gaze is more focused on a small part of the area, such as buildings and people; the other is that the distribution of the gaze point is larger, and the visual range is wider. Therefore, in the process of Longji Terraces tourism development, the visual browsing habits of tourists should be considered first, combined with the hotspots of attention, the reasonable layout of buildings, cultural symbols, etc., unified planning, prohibiting illegal construction, and highlighting the cultural symbols of Longji in the signage and other infrastructures. Firstly, the hotspots of tourists’ attention are focused on elements such as architecture or transportation, and it is necessary to promote the protection and management measures of landscape elements to maintain the satisfaction of tourists’ experience, improve the transportation situation, and enhance the attractiveness of tourism. Then, the junction of terraces and forests, buildings and mountains, mountains and sky, and other elements are of high concern and should continue to be well maintained, paying attention to the reasonable mix of vegetation and the reasonable size of the water body and other issues to maintain the coordination of the ecosystem. Secondly, tourists are more concerned about the folk activities and national characteristics of clothing and tools, should protect the authenticity of traditional folk activities, and pay attention to the inheritance of national skills, strengthen the transmission of intangible culture and interpretation of education, integrate it into the terraced rice fields, residential buildings, and other material landscape display process, so that tourists in the process of tour actively perceive the charm of the national cultural landscape and experience. Finally, tourists are more concerned about farming activities as well as agricultural tools. Traditional agricultural tools should be protected, and agricultural experience activities should be carried out to enhance the tourists’ sense of experience. On the basis of promoting the protection of terraced rice fields and rice as the core agricultural characteristics of the landscape heritage, strengthening the protection of the national cultural landscape, maintaining the overall landscape of the agricultural and cultural heritage sites, and maintaining the attractiveness of the core attractions of the destination tourism, it is very important for the agricultural Terraces of Longji Terraces to be protected and for tourism to be sustainable. Cultural heritage landscape protection and tourism sustainable development are of great significance.

② From the analysis of eye movement behavior, it can be seen that there is no significant difference in the eye fixation rate, fixation count, average fixation counts, and average fixation duration between the settlement and the architectural, ecological, ethnic cultural, and agricultural landscapes, indicating that the differences in the types of Longji Terraced Landscape are not obvious. Among the group characteristics, using permanent residence as the criterion, there were significant differences in eye fixation rate and mean fixation count; for tourists with or without the experience of revisiting the Longji Terraces, whether or not to revisit the terraces and eye fixation rate and mean fixation count yielded a significant difference, i.e., the subjects who visited the Longji Terraces for the first time had a higher eye fixation rate and mean fixation count; and using gender, ethnicity, income, age, education background, and the sampling rate as the criterion, the There was no significant difference in any of the test results. The group’s visual characteristics revealed that the subjects’ capture of the appreciation of beauty was basically convergent. Landscape type influences some eye movement behaviors, but in terms of the average fixation duration, the time tourists stay at the gaze point, and the effort of landscape cognition, still need to be adjusted. In the process of Longji Terraces tourism development, scenic spots should focus on the characteristics of different landscapes rather than over-catering to the market.

③ From the analysis of the experimental hypotheses, it can be concluded that there is no significant positive correlation between visual stimuli of the landscape and the intentional response of subjective evaluation, which indicates that the degree of information that subjects visually search for, and process does not positively affect the satisfaction of tourists. Therefore, H1 is not valid. Second, the more information is visually searched, the greater the effect on tourists’ aesthetic emotions. Therefore, landscape visual stimulation positively affects aesthetic experience, so H2 is verified. There are positive correlations between aesthetic senses, aesthetic emotions, and aesthetic spirit and intentional response, indicating that aesthetic experience has a positive influence on intentional response. Therefore, H3 is validated. Aesthetic experience does not have a mediating role between landscape visual stimuli and intentional response, and hypothesis H4 is not valid. To summarize, the model revision diagram after hypothesis testing is shown in Figure 5 below.

### 5.2. Discussion

This study investigated tourists’ visual perception, aesthetic experience, and behavioral intentions in four types of landscapes and explored the points of interest in the Longji Terraced Fields landscape as tracked by eye movement. The results found that tourists’ visual attention is influenced by different places of residence and whether they have visited before. Tourists’ gaze hot spots are concentrated at the intersections of landscape elements. The visual stimuli of the terraced fields landscape seem unable to evoke tourists’ intentions to revisit, but different eye movement parameters are somewhat related to the aesthetic experiences perceived by tourists. In addition, many scholars have explored the visual evaluation of the landscape. For example, LUO Yingshun et al. took the cultural landscape of Zhangguying Village (Zhang Guying Village) as the research object and used the SD method and eye movement analysis method to conduct experience evaluation and comprehensive statistical analysis of the cultural landscape of Zhangguying Village. It is found that the overall experience evaluation of the cultural landscape in Zhangguying Village is at a medium level [11]. CHENG Shi et al. In Nanjing, as an example, a new method for dynamic viewing and evaluation [14]. YANG Yang Taking the scenic forest in Nanjing as the research object, the principal component analysis method is adopted to transform the influence factors into a core feature index and the core factors affecting the visual quality of scenic forest [15]. From the perspective of gaze behavior and aesthetic evaluation, it was discovered that the dispersed eye movement pattern of East Asian tourists contrasts with the focal gaze of European tourists [40], which confirms the adaptability and variation of Gestalt psychology principles within a cross-cultural context [41] eye movement research methods are shifting from descriptive tools to explanatory theoretical construction, and the quantified relationship between gaze behavior and aesthetic evaluation validates the cross-cultural applicability of Gestalt psychology principles in agricultural landscapes [42].

#### 5.2.1. Practical Insights

Longji Terraced Landscape enhances the aesthetic experience of tourists by improving the aesthetic experience measures, such as enhancing the aesthetic senses of different landscapes, excavating the spiritual connotation of terraced landscapes, and so on. However, through the questionnaire and literature surveys, it is concluded that the Longji Terraced Landscape has a vague positioning, lacks attention to the aesthetic experience of different types of landscapes, and the connotation of the aesthetic experience is shown less. There is no lack of research in China that utilizes eye movement experiments to explore landscape types. For example, Hongcun in Anhui Province, through the eye movement experiment research on different landscape types conducted there, provides a reference for the research method of tourism landscapes [27]. The influence of different landscape types on aesthetic emotions is consistent with the conclusions drawn by Deng Weiwei et al. Aesthetic emotions and visual attractiveness can be considered appropriately. The influence of travelers’ visual perception on destination selection should also be considered, and photographic aesthetics should also be fully considered [35].

In this study, the mediating effect of aesthetic experience was explored, and the results of the study showed that research hypothesis H4 was not supported, and that landscape visual stimuli had a significant direct effect on intention response, not an indirect effect on intention response through aesthetic experience. This suggests that when tourists are subjected to landscape visual stimuli, it will directly and significantly affect tourists’ overall impression, comprehensive experience, satisfaction, and intention to revisit the Longji Terraced Fields landscape without going through the experience in the aesthetic process. This finding is inconsistent with Zhao Ying et al.’s conclusion that group characteristics play a mediating effect between landscape type and intention response [3]; therefore, the positive intention response of tourists to focus on the aesthetic experience of different groups of tourists, so that the aesthetic experience of the Longji Terraced Landscape can play a role in the sustainable development of the Longji Terraced Landscape, so that it can have a better impact on the overall impression of the Longji Terraced Landscape, the comprehensive experience, satisfaction and willingness to revisit the landscape to produce a better impact. When it comes to landscape visual stimulation of tourist landscapes, in the transformation process, there should be a focus on the aesthetic experience of different landscape types for tourists.

#### 5.2.2. Management Implications

The findings of the study have certain significance for the sustainable development of the Longji Terraced Landscape. The Longji Terraced Landscape has been improved through improvement measures, such as repairing characteristic buildings, folk culture inheritance, and protection of ecosystems, so as to show tourists diverse terraced landscapes. However, the positioning of the Longji Terraced Landscape is relatively single-faceted, and the whole should be integrated into diverse aesthetics and experiences, and the connotation of national culture should be fully protected. Therefore, the Longji Terraces’ national culture and settlements and architectural landscape should be shown on the basis of protection, and the wisdom of the ecological landscape should highlight the educational significance of it. In the future development of Longji Terraces, attention should be paid to the protection and development of the ecological landscape, enhancing the sensory stimulation of other landscapes, which can be enhanced from the visual, auditory, tactile, olfactory, gustatory, perceptual, and other aspects. The aesthetic emotions and spirit of Longji Terraced Landscape should be further explored so as to enhance the positive intentional responses of tourists. Through the study of aesthetic senses, suggestions can be provided to various tourism enterprises, such as hotels, restaurants, scenic spot aesthetic preferences, etc., to enhance the comprehensive feelings of tourists as well as the spiritual pursuit. Through the study of positive and negative emotions in tourism aesthetic activities, we can understand the mechanism of the formation of tourists’ euphoria, tourists’ emotional differences, and the influencing factors of tourists’ emotions, etc. In the aspect of the spirit of aesthetics, it is necessary to pay attention to aesthetic tourism’s educational value, the poetic function of literary tourism, the brand image of tourism enterprises and the construction of identity.

## Figures and Tables

**Figure 1 jemr-18-00015-f001:**
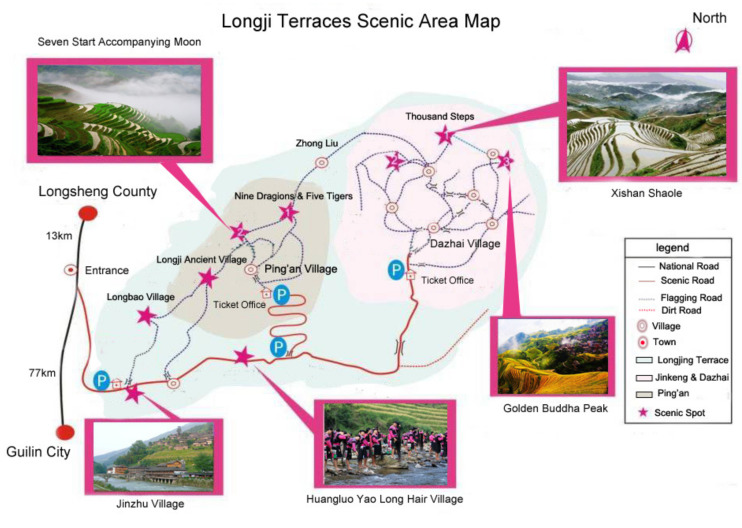
Longji Terraces Scenic Area Map.

**Figure 2 jemr-18-00015-f002:**
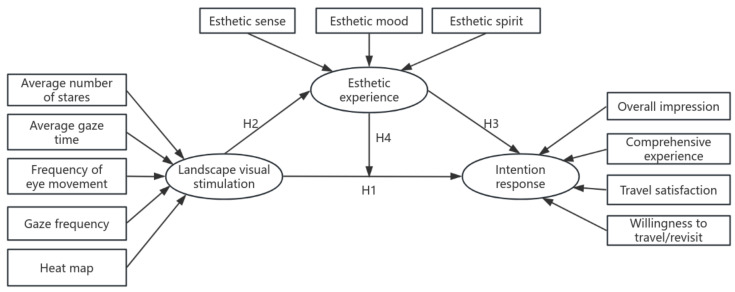
A hypothetical model of the aesthetic experience of Longji Terraced Landscape.

**Figure 3 jemr-18-00015-f003:**
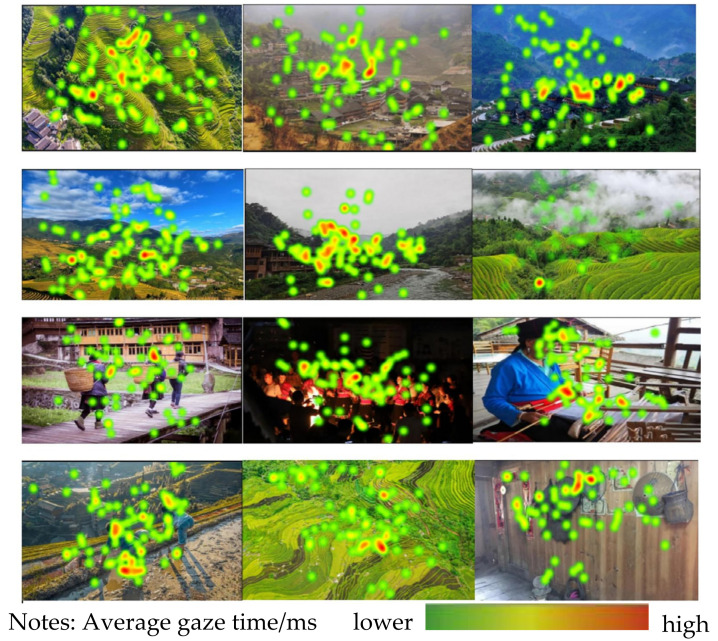
Heatmaps of four types of landscape sites in Longji Terraces.

**Figure 4 jemr-18-00015-f004:**
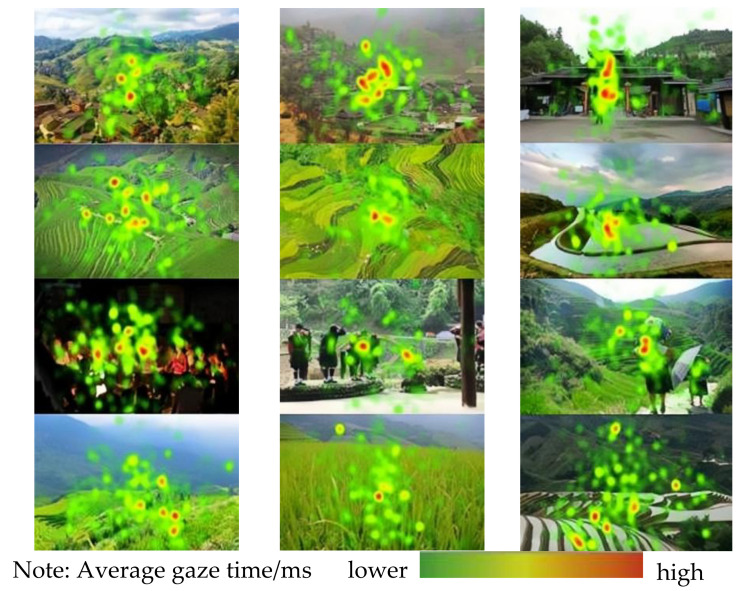
Heatmaps of four types of landscape laboratories in Longji Terraces.

**Figure 5 jemr-18-00015-f005:**
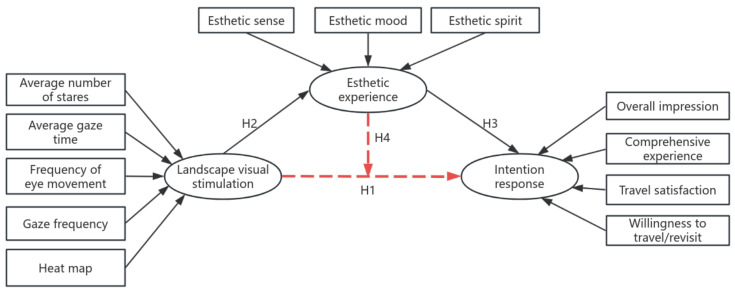
Corrected model of aesthetic experience of Longji Terraced Landscape.

**Table 1 jemr-18-00015-t001:** Correlation analysis between aesthetic experience and intentional response.

	ESE	EM	ESP	OI	CE	TS	WOT
ESE	1						
EM	0.675 **	1					
ESP	0.691 **	0.633 **	1				
OI	0.527 **	0.642 **	0.404 **	1			
CE	0.615 **	0.634 **	0.500 **	0.480 **	1		
TS	0.592 **	0.460 **	0.411 **	0.480 **	0.448 **	1	
WOT	0.209 **	0.143 **	0.182 **	0.182 **	0.184 **	0.168 **	1

** At the 0.01 level (two-tailed), the correlation is significant. Notes: ESE: esthetic senses; EM: esthetic mood; ESP: esthetic spirit; OI: overall impression; CE: comprehensive experience; TS: travel satisfaction WOT: willingness to travel/revisit.

**Table 2 jemr-18-00015-t002:** One-way ANOVA for the landscape-type field experiment.

EMI	ANOS (Times)	AGT (ms)	FOEM (Beats/s)	GF (Times/s)
Sig.	0.570	0.983	0.251	0.992
SAAL	5.52	1159.13	5.32	103.28
EL	5.91	1170.95	6.49	104.52
CL	5.48	1160.93	5.34	105.86
AL	5.25	1159.01	5.32	103.28

Notes: EMI: eye movement index; ANOS: average number of stares; AGT: average gaze time; FOEM: frequency of eye movement; GF: gaze frequency; SAAL: settlements and architectural landscapes; EL: ecological landscape; CL: cultural landscape; AL: agricultural landscape.

**Table 3 jemr-18-00015-t003:** One-way ANOVA for laboratory experiments on landscape types.

EMI	ANOS (Times)	AGT (ms)	FOEM (Beats/ms)	GF(Times/ms)
Sig.	0.001	0.749	0.684	0.288
SAAL	7.26	467.01	0.14	0.26
EL	9.38	432.85	0.16	0.30
CL	6.82	387.68	0.13	0.27
AL	6.78	439.35	0.14	0.22

Notes: EMI—eye movement index, ANOS—average number of stares, AGT—average gaze time, FOEM—frequency of eye movement, GF—gaze frequency, SAAL—settlements and architectural landscapes, EL—ecological landscape, CL—cultural landscape, AL—agricultural landscape.

**Table 4 jemr-18-00015-t004:** Correlation analysis of visual stimuli, aesthetic experience, and intentional response of field experimental landscapes.

	ESE	EM	ESP	OI	CE	TS	WOT	ANOS	AGT	FOEM	GF
ESE	1										
EM	0.245	1									
ESP	0.940 **	.25	1								
OI	0.168	0.146	.206	1							
CE	0.596 **	0.443 **	0.540 **	−0.039	1						
TS	0.386 **	0.717 **	0.373 *	0	0.372 *	1					
WOT	−0.23	0.129	−0.302 *	0.197	−0.049	0.082	1				
ANOS	−0.283	0.376*	−0.194	0.117	0.047	0.222	0.184	1			
AGT	0.054	−0.133	0.064	0.091	−0.093	−0.187	0.126	−0.169	1		
FOEM	−0.24	0.398 **	−0.167	0.122	0.081	0.267	0.23	0.985 **	−0.154	1	
GF	0.066	−0.104	0.046	0.098	−0.091	0.059	0.225	0	0.111	0.021	1

** At the 0.01 level (two-tailed), the correlation is significant. * At the 0.05 level (two-tailed), the correlation is significant. Notes: EMI—eye movement index, ANOS—average number of stares, AGT—average gaze time, FOEM—frequency of eye movement, GF—gaze frequency, ESE—esthetic senses, EM—esthetic mood, ESP—esthetic spirit, OI—overall impression, CE—comprehensive experience, TS—travel satisfaction, WOT—willingness to travel/revisit.

**Table 5 jemr-18-00015-t005:** Correlation analysis of visual stimuli, aesthetic experience, and intentional response of laboratory experimental landscapes.

	ESE	EM	ESP	OI	CE	TS	WOT	ANOS	AGT	FOEM	GF
ESE	1										
EM	0.452 **	1									
ESP	0.880 **	0.415 **	1								
OI	−0.015	−0.287	−0.028	1							
CE	−0.138	−0.157	−0.129	0.367*	1						
TS	−0.026	−0.03	−0.094	0.182	0.763 **	1					
WOT	0.397 **	0.973 **	0.352 *	−0.30 *	−0.197	−0.063	1				
ANOS	0.222	0.235	0.279	0.126	−0.073	−0.255	0.067	1			
AGT	−0.257	−0.025	−0.243	−0.067	0.197	0.206	0.124	−0.445 **	1		
FOEM	0.261	0.094	0.244	0.038	−0.169	−0.223	−0.101	0.715 **	−0.539 *	1	
GF	0.253	−0.028	0.288	0.187	−0.089	−0.216	0.066	0.772 **	−0.666 *	0.731 *	1

** At the 0.01 level (two-tailed), the correlation is significant. * At the 0.05 level (two-tailed), the correlation is significant. Notes: EMI—eye movement index, ANOS—average number of stares, AGT—average gaze time, FOEM—frequency of eye movement, GF—gaze frequency, ESE—esthetic senses, EM—esthetic mood, ESP—esthetic spirit, OI—overall impression, CE—comprehensive experience, TS—travel satisfaction, WOT—willingness to travel/revisit.

## Data Availability

The data presented in this study are available on request from the corresponding author due to privacy restrictions. The data are not publicly available to ensure compliance with participant confidentiality agreements and relevant data protection legislation.

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
