# Peer review of "Natural or Human Landscape Beauty? Quantifying Aesthetic Experience at Longji Terraces Through Eye-Tracking"

_1995-8692, 2025, doi:10.3390/jemr18030015_

Round 1

Reviewer 1 Report

Comments and Suggestions for Authors

The paper tackles the interesting subject of applying eye movement study to landscape evaluation. However, it seems that for time restriction reasons, instead of going on site, images were used. This is unacceptable since nothing compares to going on site, where the whole context can be experienced through movement not only eye movement (i.e. what is behind etc.). Otherwise the reasons of the study, especially to see if the pleasant experience leads to revisit the site, are important.

The paper is also well structured and the research is well conducted, but the initial overall hypothesis, which may be wrong, is also not spellt out before the assumptions. It is also not sufficiently clear why the assumptions are called H1-H4 so probably hypothesis, so an overall research hypothesis is subdivided. An assumption instead is something else, it shows how the model is simplified compared to the real landscape system.

There is also a problem with the title. The term "folklore beauty" does not seem adequate here. Folklore might lead to traditions but not necessarily to artificial landscape which is probably what is meant.

Another confusion of terms  is in figure 1 and respectively 4. The term "job" seems not to be adequate and maybe refers to "travel" or "experience/enjoyment".

Comments on the Quality of English Language

I hope that a more precise language and specifications on the initial problem setting of the work will help improving the paper so that it can be accepted. See the comments above.

Author Response

Comments1:The paper tackles the interesting subject of applying eye movement study to landscape evaluation. However, it seems that for time restriction reasons, instead of going on site, images were used. This is unacceptable since nothing compares to going on site, where the whole context can be experienced through movement not only eye movement (i.e. what is behind etc.). Otherwise the reasons of the study, especially to see if the pleasant experience leads to revisit the site, are important

Responses1:As this expert has pointed out, field investigations involve not only eye movements but also physical movements. However, the focus of attention varies among different tourists, and it may not be possible to ensure that they observe the same landscape environment during field investigations. Additionally, the experimental process can be influenced by surrounding environmental factors, such as noise disturbances and weather changes on the day, which introduce too many variables. Consequently, it becomes challenging to guarantee that the experimental results are reliable and scientifically valid. Therefore, this study employs images for experimentation, aiming to reduce the interference from uncontrollable factors to some extent. The concept of "pleasurable experience" refers to the emotional changes perceived by participants through observing landscape images. This paper attempts to evoke a desire in participants to visit the depicted locations by leveraging the information conveyed through landscape images.

Comments2:The paper is also well structured and the research is well conducted, but the initial overall hypothesis, which may be wrong, is also not spellt out before the assumptions. It is also not sufficiently clear why the assumptions are called H1-H4 so probably hypothesis, so an overall research hypothesis is subdivided. An assumption instead is something else, it shows how the model is simplified compared to the real landscape system.

Responses2:In this study, the SOR (Stimulus-Organism-Response) model was employed. Here, "S" refers to the stimulus, which in this context denotes the sensory stimulation elicited by the landscape; "O" originally stands for the organism, the entity that perceives the stimulus, and in this case, it refers to the tourists viewing the landscape; "R" signifies the response, which here pertains to the tourists' reactions after viewing the landscape, specifically their consideration of whether to revisit. Based on this framework, four hypotheses were proposed:

H1: Landscape visual stimuli positively influence tourists' intended responses.
H2: Landscape visual stimulation positively affects tourists' aesthetic experience.
H3: Aesthetic experience positively influences tourists' intention response.
H4: Aesthetic experience mediates between landscape visual stimuli and intended responses.

Comments3:There is also a problem with the title. The term "folklore beauty" does not seem adequate here. Folklore might lead to traditions but not necessarily to artificial landscape which is probably what is meant.

Responses3:The title has been revised to: "Natural Landscape Beauty or Human Landscape Beauty? Research on the Aesthetic Experience of the Landscape of Longji Terraces in Guilin Based on Eye-Tracking Experiments."

Comments4:Another confusion of terms  is in figure 1 and respectively 4. The term "job" seems not to be adequate and maybe refers to "travel" or "experience/enjoyment".

Responses4:As the expert mentioned, the term should refer to the sense of pleasure experienced during travel. Therefore, modifications have been made in Table 1 and Table 7, replacing "job satisfaction" with "travel satisfaction."

Reviewer 2 Report

Comments and Suggestions for Authors

It is certainly an interesting research work, based on the incorporation of a solid methodology and the authors' territorial knowledge. Some recommendations can be made:

1) Some location mapping should be presented to allow the international reader to contextualise the territory under study.

2) It is necessary to include photographs without the application of results so that the researcher himself can evaluate the results obtained.

3) It would be interesting to mention other similar studies carried out in other landscapes, whether in China, Asia or the rest of the world, in order to evaluate the results obtained by the researchers.

Good luck 

Author Response

Comments1:Some location mapping should be presented to allow the international reader to contextualise the territory under study.

Responses1:As the expert suggested, the introduction section has been expanded to include a geographical background of the Longji Terraces: The Longji Terraces, hailed as the ancestral home of terraced fields worldwide, represent a unique agricultural landscape in the Longji Mountain area of Guangxi Zhuang Autonomous Region, China, and are recognized as a significant global agricultural heritage site. Below is a detailed geographical background of the Longji Terraces: Located in Longji Town, Longsheng Various Nationalities Autonomous County, Guilin City, Guangxi Zhuang Autonomous Region, the Longji Terraces are situated between 110°04′06″ to 110°11′52″ east longitude and 25°42′33″ to 25°50′15″ north latitude. The area is one of the birthplaces of cultivated japonica rice, with evidence of primitive cultivation dating back 6,000 to 12,000 years. The terraces are part of the Yuecheng Mountains of the Nanling Range, characterized by their rolling hills, layered peaks, and majestic scenery. The Longji Terraces also exhibit a unique pattern where terraced fields and forests are interwoven; often, a stretch of terraces is followed by a forest below, a layout that effectively aids in soil and water conservation. The distinctive and rich geographical context of the Longji Terraces not only makes them a spectacular natural landscape but also an integral part of China's traditional farming culture. A map showing the location of the Longji Terraces has been included in the text.

Comments2: It is necessary to include photographs without the application of results so that the researcher himself can evaluate the results obtained.

Responses2:Original photographs and images of the subject collection site have been appended as an annex to the article.

Comments3:It would be interesting to mention other similar studies carried out in other landscapes, whether in China, Asia or the rest of the world, in order to evaluate the results obtained by the researchers.

Responses3:

  1. As the expert pointed out, additions have been made to the abstract, introduction, and other sections as follows: This study can, to some extent, enrich the objective dimensions of the visual quality assessment system. Moreover, current research on eye-tracking experiments in tourism landscapes has not yet covered terraced landscapes, providing a Chinese case study for reference in the aesthetic valuation of landscapes.

Reviewer 3 Report

Comments and Suggestions for Authors

Review

Author Response

Comments1:

Abstract  
Better describe the research methodology  - Explain the SOR model  - The results are not exact  - Lack of scientific contribution to the work  - Too much written for this type of journal  
Introduction  
On pages 1 and 2, reduce the text and specialize more specifically in the subject matter  
Supplement with at least 5 new sources to internationalize the work even more  
Define the goal of the research more clearly  

Responses1:

Abstract  Regarding the Abstract Section: Firstly, concerning the research methodology, the methods employed in this study have been more clearly delineated: 353 questionnaires were distributed on-site, and the SOR model was utilized to examine the visual stimuli and aesthetic responses perceived by tourists. Subsequently, laboratory eye-tracking was used to identify the focal points of tourists' attention on the Longji Terraces landscape. Secondly, the explanation of the SOR model is introduced in the second section on theory, and due to space constraints, no further detailed elaboration is provided here. The research findings have been refined: different residential locations and revisit statuses affect tourists' visual attention, with the most attention paid to the intersections of landscape elements. Moreover, although landscape visual stimuli do not significantly influence intentional responses, eye-tracking parameters are positively correlated with aesthetic experiences. Lastly, regarding the scientific contribution, this study enriches the objective dimensions of the visual quality assessment system to some extent. Additionally, current research on eye-tracking experiments in tourism landscapes has not yet covered terraced landscapes, providing a referenceable Chinese case for the aesthetic value of landscapes.

Introduction In the Introduction Section, references not strongly related to the article's theme have been removed, and the latest research findings strongly related to the article's study have been added. For example, Yu Gao et al. used eye-tracking technology to analyze the visual behavior characteristics and differences in different types of forest landscape spaces; Minanshu Sinha et al. explored how the gender selection of fast-moving consumer goods (FMCG) advertising endorsers can maximize advertising effectiveness; Yiping Liu et al. used eye-tracking to evaluate the audio-visual interaction effects of forest landscapes, finding that different types of forest landscapes affect tourists' perceptions; Xingcan Zhou et al. used eye-tracking technology to explore visual behavior differences related to landscape elements in urban lakeside parks, finding more attention paid to artificial elements; Chang Li et al. measured seven levels of forest light exposure through virtual reality eye-tracking technology to explore the impact of different brightness levels on forest visual perception, thereby enhancing the international level of this research. In this process, the research objectives of this study have been more clearly understood: attempting to reflect participants' objective perceptions through eye-tracking experiments to enrich the objective dimensions of the visual quality assessment system. Furthermore, current research on eye-tracking experiments in tourism landscapes has not yet involved terraced landscapes, so this study can provide a Chinese case for the objective evaluation of visual perception in terraced landscapes.

Comments2:
Theory and Model Assumptions  
Try to shorten this chapter, leave the most important facts  
Figure 1. improve the result  
On page 4, under hypothesis 1, there is accompanying text, but under h2, h3, h4 it is not there? 

Responses2:In the Theory and Research Hypotheses Section, firstly, due to article length constraints, the related theories involved in the research have been simplified, retaining only the theories that best explain this study. Secondly, regarding the ambiguity in the textual explanation of the research hypotheses, the hypotheses corresponding to existing research findings have been subdivided in this revision, retaining the most relevant and important research outcomes.

Comments3:Research methods  
A chapter that is significantly summarized in this form is quite confusing and difficult to read 

Responses3:In the Research Methodology Section, this part has been reorganized to include five sections: research subjects, preliminary research preparation, research process, experimental data processing, and data analysis methods. The preliminary preparation includes: experimental landscape images, subjects, experimental equipment, questionnaire scales, and the selection of eye-tracking indicators, making the article clearer and more understandable.

Comments4:Results  
Also summarize and make the text and images more clear and readable  

Responses4:In the Research Results Section, starting from the reliability analysis of the data, to the correlation levels of the measurement dimensions, and finally to the analysis results of the eye-tracking experiments. The results of the eye-tracking experiments include a comparative analysis of visual attention based on eye-tracking heatmaps, eye-tracking analysis based on different landscape types, and eye-tracking analysis based on different demographic characteristics, followed by model hypotheses. The comparative analysis of visual attention based on eye-tracking heatmaps and the eye-tracking analysis based on different landscape types can respectively reveal which parts of the landscape and which types of landscapes tourists are more interested in. The eye-tracking analysis based on demographic characteristics shows that different residential locations and revisit statuses affect tourists' visual attention, forming a coherent structure.

Comments5:Discussion  
There is a lack of external sources for comparison, I suggest introducing a conclusion where it 
would be briefly and clearly answered the objective of the paper 

Responses5:In the Discussion Section, firstly, the fifth part includes the conclusion and discussion. Following the discussion in section 5.2, the research objectives and findings of this study have been added: this study investigated tourists' visual perceptions, aesthetic experiences, and behavioral intentions across four landscape types, and explored the focal points of tourists' attention on the Longji Terraces landscape using eye-tracking. The findings reveal that different residential locations and revisit statuses affect tourists' visual attention, with tourists' gaze hotspots concentrated at the intersections of landscape elements. The visual stimuli of the terraced landscape do not seem to inspire tourists' revisit intentions, but different eye-tracking parameters are somewhat linked to the aesthetic experiences felt by tourists. 

Comments6:Literature  
Add external sources to internationalize the paper 

Responses6:The comparative analysis with foreign literature related to this study has been supplemented and refined in conjunction with the latest domestic and international research literature.

Reviewer 4 Report

Comments and Suggestions for Authors

The submitted manuscript deals with the modeling of aesthetic landscape constructions. The foundations of this are very conditional. This is where the central weakness of the article becomes apparent: it is not made clear in a differentiated way what is meant by aesthetics, what is meant by landscape or by landscape aesthetics. What is undertaken is a pure preference analysis. 
If the aim of the essay is to deal with landscape aesthetics, it seems unavoidable to deal with aesthetics as a philosophical discipline, the construction of landscape and the construction of landscape aesthetics. Otherwise, it seems necessary to explain one's own scientific-theoretical point of view in order to make it clear that a differentiated examination of the underlying concepts does not take place. 
Regardless of this, the cultural boundness of aesthetic judgments and the concept of landscape, which cannot simply be assumed to be ubiquitous, must be worked out. 
In this respect, a thorough revision and justification of one's own understanding and approach seems necessary.

Author Response

Comment:The submitted manuscript deals with the modeling of aesthetic landscape constructions. The foundations of this are very conditional. This is where the central weakness of the article becomes apparent: it is not made clear in a differentiated way what is meant by aesthetics, what is meant by landscape or by landscape aesthetics. What is undertaken is a pure preference analysis. 
If the aim of the essay is to deal with landscape aesthetics, it seems unavoidable to deal with aesthetics as a philosophical discipline, the construction of landscape and the construction of landscape aesthetics. Otherwise, it seems necessary to explain one's own scientific-theoretical point of view in order to make it clear that a differentiated examination of the underlying concepts does not take place. 
Regardless of this, the cultural boundness of aesthetic judgments and the concept of landscape, which cannot simply be assumed to be ubiquitous, must be worked out. 
In this respect, a thorough revision and justification of one's own understanding and approach seems necessary.

Response:Regarding the issue of modeling aesthetic landscape construction raised by the expert, I would like to clarify that this is not the primary focus of our study. In this research, we approach the topic from the perspective of the intersection between aesthetics and tourism studies. Tourism aesthetics should be considered an interdisciplinary field that combines aesthetics and tourism, examining the phenomena and principles of aesthetics within tourism activities. This study emphasizes the integration of environmental aesthetics and tourism experiences, treating tourists as participants in the environment. We explore how tourists' aesthetic experiences are stimulated in different types of environments, particularly in the context of terraced landscapes, and attempt to further analyze which types of terraced landscapes or which specific parts of these landscapes attract more interest during the process of aesthetic experience generation through eye-tracking experiments. We selected the Longji Terraces as a representative case study for this purpose. In this process, considerations regarding post-experience landscape construction have been somewhat lacking. As the expert pointed out, the current research focus is indeed a form of preference analysis. The issue of subsequent landscape construction is something I intend to explore more deeply in future research. The ultimate goal of tourism aesthetics is to achieve "poetic dwelling" (Heidegger) – reconciling humans with nature, tradition with modernity, and the self with the other through aesthetic experiences.

Round 2

Reviewer 1 Report

Comments and Suggestions for Authors

Many thanks, the paper improved. Thank you are for the explanations. Responses 2-4 are fully acceptable. With regard to response 1, I suggest to more precisely spell out in the paper that it refers to the desire to become a tourist in the given area, i.e. prospective tourists respectively maybe to tourist memories. And I suggest to also add a comment if these images were accompanied by a description or contained only the visual material.

Author Response

Comment

:Many thanks, the paper improved. Thank you are for the explanations. Responses 2-4 are fully acceptable. With regard to response 1, I suggest to more precisely spell out in the paper that it refers to the desire to become a tourist in the given area, i.e. prospective tourists respectively maybe to tourist memories. And I suggest to also add a comment if these images were accompanied by a description or contained only the visual material.

Response:

Regarding the expert's precise statement about "stimulating tourists to become potential visitors to the site":
We concur with this observation and have refined the corresponding description in the introduction section. As specified in the concluding sentence on page 4 of the manuscript: "provides reference significance for better attracting potential tourists to the terrace field scenic area."

Regarding the expert's inquiry about image captions:
This study incorporates two categories of visual data:

  1. Eye-tracking heatmaps of four landscape types

  2. Thermal imaging maps of four landscape types

The color gradation in both image types indicates average fixation duration, where:

  • Warmer hues (red spectrum) represent longer gaze durations

  • Cooler hues (green spectrum) indicate shorter gaze durations
    This chromatic scale convention is explicitly explained in the captions of Figure 2 and Figure 3 on page 11 of the manuscript, beneath the respective images.

Reviewer 3 Report

Comments and Suggestions for Authors

The work was corrected according to the instructions. I have no objections.

Author Response

comment:The work was corrected according to the instructions. I have no objections.

response:Thank you for your positive feedback and thorough review of our work. We are pleased to hear that the revisions meet your approval. Please let us know if any further clarifications are needed.

Reviewer 4 Report

Comments and Suggestions for Authors

Dear authors, if you want the context of aesthetics in its philosophical, social-scientific, even spatial-scientific differentiation (the same applies to landscape), then you should not use these terms either, as you cannot do justice to the 'great semantic court'. In this respect, you should speak of 'preference' instead of 'aesthetics' and use the more neutral term 'space' instead of 'landscape'.

Author Response

Comment:Dear authors, if you want the context of aesthetics in its philosophical, social-scientific, even spatial-scientific differentiation (the same applies to landscape), then you should not use these terms either, as you cannot do justice to the 'great semantic court'. In this respect, you should speak of 'preference' instead of 'aesthetics' and use the more neutral term 'space' instead of 'landscape'.

Response:

First and foremost, we would like to express our gratitude to the experts for their valuable comments and suggestions. Our responses to these observations are as follows:

Regarding the perceived inaccuracy of "aesthetic" usage:
Based on the theoretical framework presented in "Theoretical Considerations on Semantic Categories" by scholars Shao Jingmin and Zhao Chunli, published in "World Chinese Teaching", semantic categories essentially constitute grammatical categories derived from grammatical meaning. There exists no formless meaning nor meaningless form in linguistic study. The artificial separation of these elements represents a metaphysical approach that essentially negates the objective existence of Chinese grammar. In our research context, the term "aesthetic" does not stand alone but forms the compound concept of "aesthetic experience," which should be interpreted holistically. This term specifically captures visitors' experiential engagement when appreciating terrace landscapes, emphasizing their subjective perception. The alternative term "preference" merely denotes partial favorability without adequately conveying the nuanced personal experience. Given this semantic imprecision, we maintain that "aesthetic" remains the more appropriate choice.

Concerning the perceived lack of neutrality in "landscape" usage:  
The term "landscape" typically denotes a visually integrated manifestation of a geographical area, encompassing both natural scenery and artificial constructs. In geographical contexts, it may refer to comprehensive regional characteristics spanning natural, economic, and cultural dimensions. Conversely, "space" emphasizes physical attributes, geometric properties, and functional utilization—examining spatial qualities and structural dynamics in spatial sciences, or exploring socio-spatial interactions in sociological studies. Our research investigates whether variations in *landscape typology* influence visitors' aesthetic experiences, focusing on comprehensive experiential transformations rather than mere spatial alterations. Accordingly, we have defined four landscape categories:  
1. Settlement and Architectural Landscape**: Composite formations of buildings, infrastructure, and surrounding environments  
2. Ecological Landscape: Comprising terraces, mountains, and water systems  
3. Ethnocultural Landscape: Manifesting ethnic characteristics through festivals, attire, and local customs  
4. Agricultural Landscape: Shaped by farming practices  

This categorical framework substantiates our rationale for employing "landscape" as the most contextually appropriate term.